# Antibody Binding to SARS-CoV-2 S Glycoprotein Correlates with but Does Not Predict Neutralization

**DOI:** 10.3390/v12111214

**Published:** 2020-10-26

**Authors:** Shilei Ding, Annemarie Laumaea, Mehdi Benlarbi, Guillaume Beaudoin-Bussières, Romain Gasser, Halima Medjahed, Marie Pancera, Leonidas Stamatatos, Andrew T. McGuire, Renée Bazin, Andrés Finzi

**Affiliations:** 1Centre de Recherche du CHUM, Montréal, QC H2X 0A9, Canada; ding.shilei@gmail.com (S.D.); annemarie.laumaea@umontreal.ca (A.L.); mbenl068@uottawa.ca (M.B.); guillaume.beaudoin-bussieres@umontreal.ca (G.B.-B.); romain.gasser@umontreal.ca (R.G.); halima.medjahed.chum@ssss.gouv.qc.ca (H.M.); 2Département de Microbiologie, Infectiologie et Immunologie, Université de Montréal, Montreal, QC H2X 0A9, Canada; 3Fred Hutchinson Cancer Research Center, Vaccines and Infectious Diseases Division, Seattle, 98109 WA, USA; mpancera@fredhutch.org (M.P.); lstamata@fredhutch.org (L.S.); amcguire@fredhutch.org (A.T.M.); 4Vaccine Research Center, National Institutes of Allergy and Infectious Diseases, National Institute of Health, Bethesda, 20892 MD, USA; 5Department of Global Health, University of Washington, Seattle, 98195 WA, USA; 6Héma-Québec, Affaires Médicales et Innovation, Québec, QC G1V 5C3, Canada; Renee.Bazin@hema-quebec.qc.ca; 7Department of Microbiology and Immunology, McGill University, Montreal, QC H3A 2B4, Canada

**Keywords:** COVID-19, SARS-COV-2, convalescent plasma, ELISA, neutralization, virus capture assay

## Abstract

Convalescent plasma from SARS-CoV-2 infected individuals and monoclonal antibodies were shown to potently neutralize viral and pseudoviral particles carrying the S glycoprotein. However, a non-negligent proportion of plasma samples from infected individuals, as well as S-specific monoclonal antibodies, were reported to be non-neutralizing despite efficient interaction with the S glycoprotein in different biochemical assays using soluble recombinant forms of S or when expressed at the cell surface. How neutralization relates to the binding of S glycoprotein in the context of viral particles remains to be established. Here, we developed a pseudovirus capture assay (VCA) to measure the capacity of plasma samples or antibodies immobilized on ELISA plates to bind to membrane-bound S glycoproteins from SARS-CoV-2 expressed at the surface of lentiviral particles. By performing VCA, ELISA, and neutralization assays, we observed a strong correlation between these parameters. However, while we found that plasma samples unable to capture viral particles did not neutralize, capture did not guarantee neutralization, indicating that the capacity of antibodies to bind to the S glycoprotein at the surface of pseudoviral particles is required but not sufficient to mediate neutralization. Altogether, our results highlight the importance of better understanding the inactivation of S by plasma and neutralizing antibodies.

## 1. Introduction

The etiologic agent of the coronavirus disease (COVID-19), the severe acute respiratory syndrome coronavirus 2 (SARS-CoV-2), spread from China to the rest of the world and is the cause of the current pandemic [1]. In the absence of an effective vaccine to prevent SARS-CoV-2 infection, alternative approaches to treat or prevent acute COVID-19 are desperately needed. A promising approach is the use of convalescent plasma containing anti-SARS-CoV-2 antibodies collected from donors who have recovered from COVID-19 [2]. Convalescent plasma therapy has been successfully used in the treatment of SARS, Middle East respiratory syndrome (MERS), and influenza H1N1 pandemics and is associated with the improvement of clinical outcomes [3,4,5]. The transfer of convalescent plasma to COVID-19 patients is shown to be well-tolerated and has presented some positive signs, such as viral load reduction and viremia disappearance in treated severe patients [6,7,8,9,10]. Similarly, some antibodies targeting the spike (S) glycoprotein from SARS-CoV-2, isolated from virus-infected individuals, were shown to potently neutralize viral and pseudoviral particles carrying the S glycoprotein of the virus [11,12,13,14,15,16] and, in some instances, have protected small animals from SARS-CoV-2 infection [17,18].

Nonetheless, a non-negligent proportion of plasma samples from infected individuals as well as S-specific monoclonal antibodies are reported to be non-neutralizing despite efficient interaction with the S glycoprotein in different biochemical assays using soluble recombinant forms of S or when expressed at the cell surface [12,16], thus raising the question of how neutralization relates to binding of S in the context of viral or pseudoviral particles. To address this question, we adapted a previously-described virus capture assay (VCA) [19] to measure the capacity of plasma samples and monoclonal antibodies to bind to membrane-bound S glycoproteins from SARS-CoV-2 expressed at the surface of lentiviral particles and compared to their neutralization activity.

## 2. Materials and Methods

### 2.1. Ethics Statement

All subjects gave their informed consent for inclusion before they participated in the study. The study was conducted in accordance with the Declaration of Helsinki, and the protocol was approved by the Ethics Committee of CHUM (19.381, approved on 25 March 2020). Donors met all donor eligibility criteria: previous confirmed COVID-19 infection and complete resolution of symptoms for at least 14 days.

### 2.2. Plasmids

The plasmids expressing the human coronavirus spike of SARS-CoV-2 and SARS-CoV-1 were kindly provided by Stefan Pöhlman, as previously reported [20]. The pNL4.3 Nef- Luc Env- was obtained from the NIH AIDS Reagent Program. The vesicular stomatitis virus G (VSV-G)-encoding plasmid (pSVCMV-IN-VSV-G) was previously described [14].

### 2.3. Cell Lines

293T human embryonic kidney cells (HEK293T) (obtained from American Type Culture Collection (ATCC), Manassas, VA, USA) and Cf2Th cells (a kind gift from Joseph Sodroski, Dana Farber Cancer Institute (DFCI), Boston, MA., USA. ) were maintained at 37 °C under 5% CO_2_ in Dulbecco’s modified Eagle’s medium (DMEM) (Wisent, Saint-Jean-Baptiste, QC, Canada) containing 5% fetal bovine serum (VWR, Radnor, PA, USA), 100 UI/mL of penicillin, and 100 μg/mL of streptomycin (Wisent). The 293T-ACE2 cell line was previously reported [14].

### 2.4. ELISA

The SARS-CoV-2 receptor binding domain (RBD) ELISA assay used was recently described [14]. Briefly, recombinant SARS-CoV-2 S RBD (2.5 μg/mL), or bovine serum albumin (BSA) (2.5 μg/mL) as a negative control, were prepared in PBS and were adsorbed to plates (MaxiSorp, Nunc, Thermo Fisher Scientific, Waltham, MA, USA.) overnight at 4 °C. Coated wells were subsequently blocked with blocking buffer (Tris-buffered saline (TBS) containing 0.1% Tween20 and 2% bovine serum albumin (BSA)) for 1 hour at room temperature. Wells were then washed four times with washing buffer (Tris-buffered saline (TBS) containing 0.1% Tween20). CR3022 monoclonal antibody (mAb) (50 ng/mL) or serial dilutions of plasma from SARS-CoV-2-infected or uninfected donors (1/100; 1/250; 1/500; 1/1000; 1/2000; 1/4000) were prepared in a diluted solution of blocking buffer (0.1% BSA) and incubated with the RBD-coated wells for 90 min at room temperature. Plates were washed four times with washing buffer, followed by incubation with Horseradish peroxidase (HRP) labeled secondary antibodies (Abs) (diluted in a diluted solution of blocking buffer (0.4% BSA)) for 1 h at room temperature, followed by four washes. HRP enzyme activity was determined after the addition of a 1:1 mix of Western lightning oxidizing and luminol reagents (PerkinElmer, Waltham, MA, USA.). Light emission was measured with an LB941 TriStar luminometer (Berthold Technologies, Bad Wildbad, Germany). Signal obtained with BSA was subtracted for each plasma sample and was then normalized to the signal obtained with CR3022 mAb present in each plate.

### 2.5. Virus Capture Assay

The assay was modified from a previously published method [19]. Briefly, pseudoviral particles were produced by transfecting 2 × 10^6^ HEK293T cells with pNL4.3 Nef- Luc Env- (3.5 μg), pSVCMV-IN-VSV-G (1μg) and plasmids encoding for SARS-CoV-1 or SARS-CoV-2 spike (3.5 μg) glycoproteins using the standard calcium phosphate protocol. 48 hours later, viron-containing supernatants were collected, and the cell debris was removed through centrifugation (486× *g* for 10 min). To immobilize plasma on ELISA plates, white MaxiSorp ELISA plates (Thermo Fisher Scientific, Waltham, MA, USA.) were incubated with 1:500 diluted plasma in 100 μL phosphate-buffered saline (PBS) overnight at 4 °C. Unbound antibodies or plasma were removed by washing the plates twice with PBS. Plates were subsequently blocked with 3% BSA in PBS for 1 h at room temperature. After two washes with PBS, 200 μL of virus-containing supernatant was added to the wells. Viral capture by any given plasma sample was visualized by adding 1 × 10^4^ SARS-CoV-2-resistant Cf2Th cells in full DMEM medium per well. Forty-eight hours post-infection, cells were lysed by the addition of 30 μL of passive lysis buffer (Promega, Madison, WI, USA.) and three freeze-thaw cycles. An LB941 TriStar luminometer (Berthold Technologies) was used to measure the luciferase activity of each well after the addition of 100 μL of luciferin buffer (15 mM MgSO_4_, 15 mM KH_2_PO_4_ (pH 7.8), 1 mM ATP, and 1 mM dithiothreitol) and 50 μL of 1 mM D-luciferin potassium salt (Prolume, Randolph, VT, USA.).

### 2.6. Virus Neutralization Assay

Target cells were infected with single-round luciferase-expressing lentiviral particles, as described previously [14]. Briefly, 293T cells were transfected by the calcium phosphate method with pNL4.3 Nef- Luc Env- and a plasmid encoding for SARS-CoV-2 spike at a ratio of 5:4. Two days post-transfection, viron-containing supernatants were harvested and stored at −80 °C until use. 293T-ACE2 target cells were seeded at a density of 1 × 10^4^ cells/well in 96-well luminometer-compatible tissue culture plates (PerkinElmer) 24 hours before infection. Recombinant viruses in a final volume of 100 μL were incubated with the indicated plasma dilutions (1/50; 1/250; 1/1250; 1/6250; 1/31,250) for 1 h at 37 °C and were then added to the target cells, followed by incubation for 48 hours at 37 °C; cells were lysed by the addition of 30 μL of passive lysis buffer (Promega), followed by one freeze-thaw cycle. An LB941 TriStar luminometer (Berthold Technologies) was used to measure the luciferase activity of each well after the addition of 100 μL of luciferin buffer (15 mM MgSO_4_, 15 mM KPO_4_ (pH 7.8), 1 mM ATP, and 1 mM dithiothreitol) and 50 μL of 1 mM D-luciferin potassium salt (Prolume). The neutralization half-maximal inhibitory dilution (ID_50_) represents the plasma dilution required to inhibit 50% of the infection of 293T-ACE2 cells by the pseudoviral particles.

## 3. Results and Discussion

### 3.1. Virus Capture Assay

To measure the capacity of plasma samples and monoclonal antibodies immobilized on ELISA plates to bind to membrane-bound S glycoproteins from SARS-CoV-2 expressed at the surface of lentiviral particles, we adapted a previously described virus capture assay (VCA) [19]. The pseudoviral particles used in this assay were generated by transfecting HEK293T cell with the pNL4.3 Nef- Luc Env- construct [21,22,23,24]. This construct was co-transfected with a plasmid encoding the S glycoprotein from SARS-CoV-2 and a plasmid encoding the G glycoprotein from vesicular stomatitis virus (VSV-G), resulting in a virus capable of a single round of infection. Pseudovirus-containing supernatants were added to plasma or antibody-coated ELISA plates, and unbound pseudoviruses were washed away. Retention of pseudoviruses on the surface of the plate by S-specific antibodies was visualized by the addition of Cf2Th cells. These cells were refractory to SARS-CoV-2 S-mediated entry (not shown) but were infected by the bound pseudoparticles via the incorporated G glycoprotein from VSV. Cf2Th infection was measured as a function of luciferase activity two days later. A scheme of the assay is depicted in Figure 1A. This assay was used with plasma samples from SARS-CoV-2-convalescent individuals and, as expected, showed a robust capture of pseudoviral particles, whereas plasma obtained from SARS-CoV-2 uninfected individuals failed to do so (Figure 1B). Similarly, previously described SARS-CoV-2 S-specific monoclonal antibodies isolated from convalescent donors [12] were able to capture pseudoviruses expressing the S glycoprotein from SARS-CoV-2. Of note, the capture was specific since plasma from convalescent individuals and monoclonal antibodies failed to capture similar pseudoviruses expressing the S glycoprotein from SARS-CoV-1 (Figure 1B).

### 3.2. Recognition of S Glycoproteins at the Surface of Pseudoviral Particles is Required but not Sufficient to Neutralize

It is presently unclear whether the capacity of plasma from convalescent donors or monoclonal antibodies to neutralize pseudoviral particles correlates with their capture efficiency. Therefore, we first performed anti-SARS-CoV-2 RBD IgG ELISA using plasma samples recovered six weeks after symptoms onset from twenty-five convalescent individuals, as described [13,14]. To compare the different binding capacities, relative light units (RLU) obtained with the BSA (negative control) were subtracted and further normalized to the signal obtained with the anti-RBD CR3022 mAb present in each plate, as shown in Figure 2A, and the area under the curve (AUC) was calculated based on RLU. Compared to the plasma from healthy donors, we noticed that most convalescent plasma samples contained SARS-CoV-2 RBD specific IgG. As expected, we noticed a large heterogeneity with respect to the amount of anti-RBD IgG antibodies among the different samples. We then used the same samples to evaluate their neutralization potential of pseudoparticles bearing the SARS-CoV-2 S glycoprotein using 293T cells stably expressing ACE2 as target cells, as described [14]. Neutralizing activity, as defined by the neutralization half-maximal inhibitory dilution (ID_50_), is shown in Figure 2B. Similar to the variable levels of anti-RBD IgG detected in these samples (Figure 2A), these results illustrate the variable capacity of different convalescent plasma samples to neutralize. We then measured the capacity of the same plasma samples to capture viral particles. As mentioned above, HEK293T cells were co-transfected with pNL4.3 Nef- Luc Env- together with plasmids encoding the SARS-CoV-2 S glycoprotein and VSV-G; released viral particles were collected two days after transfection. For the VCA, 96-well microplates were coated with plasma recovered from SARS-CoV-2 uninfected individuals (control) or plasma from convalescent donors or with the receptor-binding domain (RBD)-specific CR3022 monoclonal antibody. This antibody was added to each plate for normalization purposes. Pseudoviral particles were added to the plates and incubated for 4 hours at 37 °C; the plates were then washed to remove unbound pseudoviruses. Cf2Th cells were added to the wells, incubated at 37 °C, and lysed 48 hours later to measure luciferase activity. As reported in Figure 2C, while a few plasma samples from convalescent donors failed to capture viral particles, similar to plasma from uninfected donors, most did. As previously reported [13,14], we observed a very significant correlation between anti-RBD IgG detected by ELISA and neutralization (Figure 2D). Interestingly, even though we observed a significant correlation between virus capture and neutralization (Figure 2E), virus capture did not always translate into neutralization. Indeed, we observed several convalescent plasma samples that were able to efficiently capture pseudoviral particles but did not neutralize, thus indicating that while the interaction of SARS-CoV-2 S glycoprotein is required for virus neutralization, it is not sufficient *per se*. Of note, we observed a strong correlation between the presence of RBD-targeting antibodies and VCA, suggesting that detection of the receptor-binding domain by antibodies present in the plasma contributes to viral capture (Figure 2F).

### 3.3. The Capacity of Convalescent Plasma to Bind to the S Glycoprotein of SARS-CoV-2 and Neutralize Pseudoviral Particles Decreases Over Time

Several reports have described a significant decrease in the neutralization capacity of plasma from convalescent individuals starting six weeks after symptoms onset [13,14,25,26]. To evaluate whether this was related to the capacity of plasma to recognize the S glycoprotein, we analyzed the level of anti-SARS-CoV-2 RBD IgG, the neutralization, and viral capture capacity of serological samples obtained from fifteen convalescent donors at six and ten weeks after symptoms onset. As previously reported [13,14], we observed a significant decrease in the level of anti-SARS-CoV-2 RBD IgG and the neutralizing capacity of samples over time (Figure 3A,B). A similar decrease in their capacity to capture viral particles was observed in the present study (Figure 3C). Accordingly, we noted strong correlations between neutralization and the level of anti-SARS-CoV-2 RBD IgG (Figure 3D), between neutralization and virus capture (Figure 3E), and between the level of anti-SARS-CoV-2 RBD IgG and virus capture (Figure 3F), thus suggesting that the decrease in neutralization over time might be due to the disappearance of antibodies able to recognize the S glycoprotein at the surface of viral particles.

Altogether, using a newly designed virus capture assay, here we reported that the capacity of antibodies to bind to the S glycoprotein at the surface of viral particles or to the RBD domain is required but not sufficient to mediate neutralization. Efforts to better understand the link between antibody interaction with S but also other viral proteins present on authentic SARS-CoV-2 viral particles, such as M and E, and virus neutralization might assist ongoing vaccine efforts aimed at eliciting neutralizing antibodies.

## Figures and Tables

**Figure 1 viruses-12-01214-f001:**
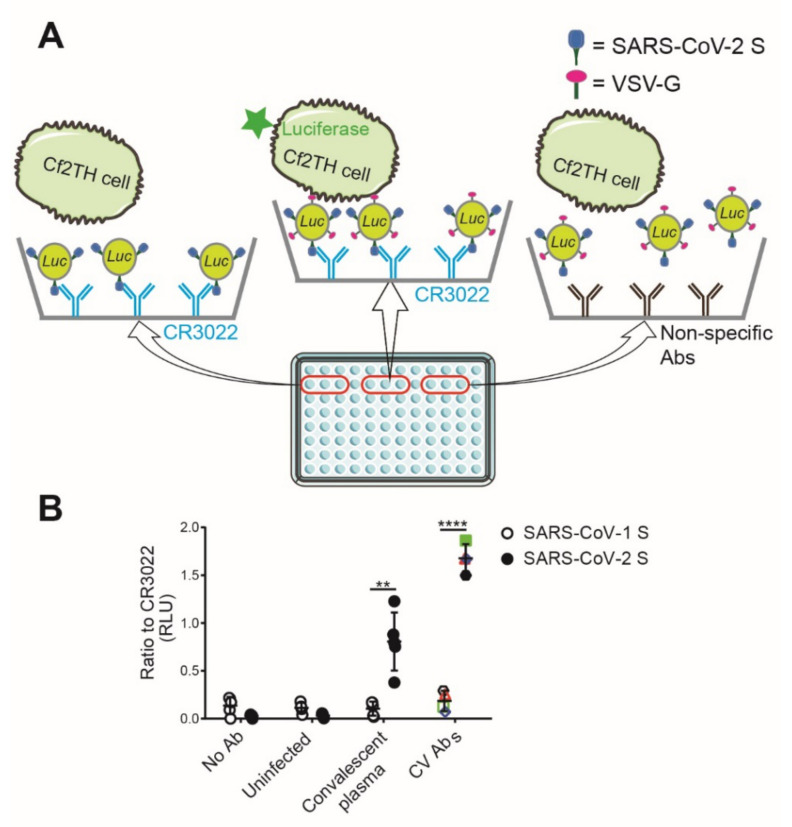
Depiction of the virus capture assay. (**A**) As shown in the scheme, 96-well ELISA plates were coated with SARS-CoV-2 S specific monoclonal antibodies or plasma recovered from SARS-CoV-2 uninfected individuals (uninfected) or plasma recovered six weeks after symptoms onset (convalescent plasma). Viral particles encoding for luciferase and bearing VSV-G glycoprotein and SARS-CoV-1 S or SARS-CoV-2 S glycoproteins were added to the wells. Free virions were washed away, and Cf2Th cells, which are refractory to SARS-CoV-2 S-mediated entry, were added to the wells. After 48 hours, cells were lysed, and the luciferase activity was measured. (**B**) Previously described SARS-CoV-2 S specific monoclonal antibodies [12] (CV1—black hexagon, CV2—red triangle, CV24—green square, and CV30—blue diamond), plasma recovered from SARS-CoV-2 uninfected individuals (uninfected), or plasma recovered six weeks after symptoms onset (convalescent plasma) were tested for the binding with viral particles bearing SARS-CoV-1 S (hollow) or SARS-CoV-2 S glycoproteins (solid). The relative light unit (RLU) obtained from CR3022 was used as control (set as one). Data shown are the mean ± standard deviation (SD) of three independent experiments performed in triplicate. Statistical significance was evaluated using a paired t-test (**, *p* < 0.01, ****, *p* < 0.0001).

**Figure 2 viruses-12-01214-f002:**
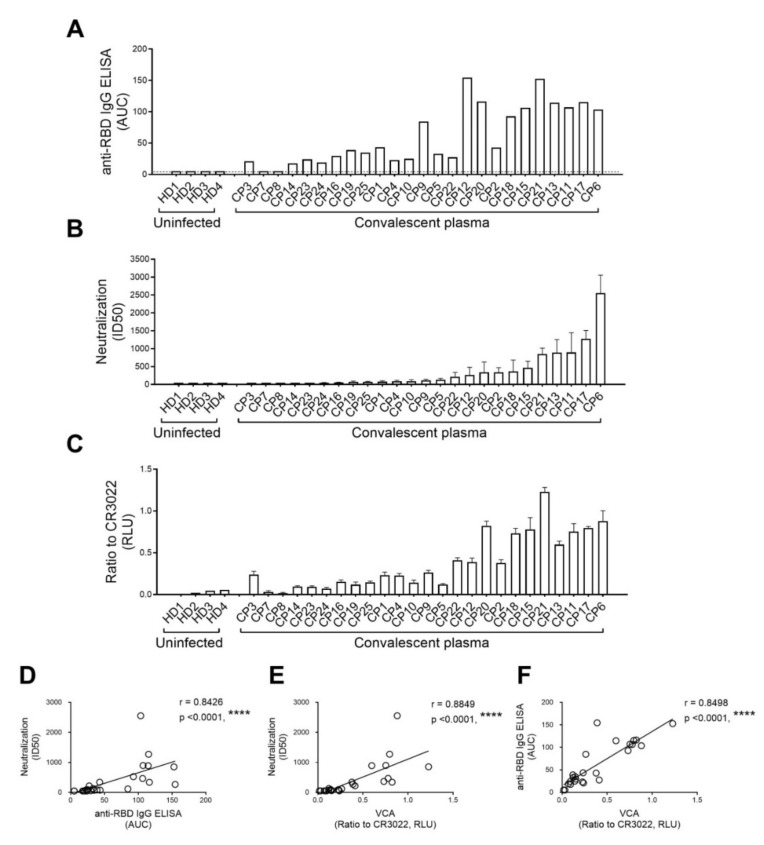
Recognition of S glycoproteins at the surface of pseudoviral particles is required but not sufficient to neutralize. Plasma samples recovered from SARS-CoV-2 uninfected individuals (uninfected) or plasma recovered six weeks after symptoms onset (convalescent plasma) were used for (**A**) anti-SARS-CoV-2 RBD IgG ELISA, (**B**) neutralization against pseudoviral particles expressing the SARS-CoV-2 S glycoprotein, and (**C**) virus capture assay (VCA). (**A**) Anti-RBD antibody binding was detected using anti-IgG-HRP. Relative light units (RLU) obtained with BSA (negative control) were subtracted and further normalized to the signal obtained with the anti-RBD CR3022 mAb present in each plate. The areas under the curve (AUC) were calculated based on RLU using GraphPad Prism software. AUC below threshold (dashed line) was considered no binding. (**B**) Pseudoviral particles coding for the luciferase reporter gene and bearing the SARS-CoV-2 S glycoprotein were used to infect 293T-ACE2 cells. Pseudoviruses were incubated with serial dilutions of plasma at 37 °C for 1 hour prior to infection of 293T-ACE2 cells. Neutralization half-maximal inhibitory plasma dilution (ID_50_) was determined using a normalized non-linear regression using Graphpad Prism software. Data shown are the mean ± SD of three independent experiments performed in triplicate. (**C**) VSV-G-pseudotyped viral particles expressing the SARS-CoV-2 S glycoprotein were added to plates coated with plasma samples. Free virions were washed away, and Cf2Th cells were added to the wells. After 48 hours, cells were lysed, and the luciferase activity was measured. Luciferase signals were normalized to those obtained with the RBD-specific CR3022 antibody. Data shown are the mean ± SD of three independent experiments performed in triplicate. (**D**) Correlations between neutralization potency and anti-RBD IgG ELISA, (**E**) neutralization and VCA, and (**F**) anti-RBD IgG ELISA and VCA were calculated. Statistical significance was established with the Spearman rank correlation test (****, *p* < 0.0001).

**Figure 3 viruses-12-01214-f003:**
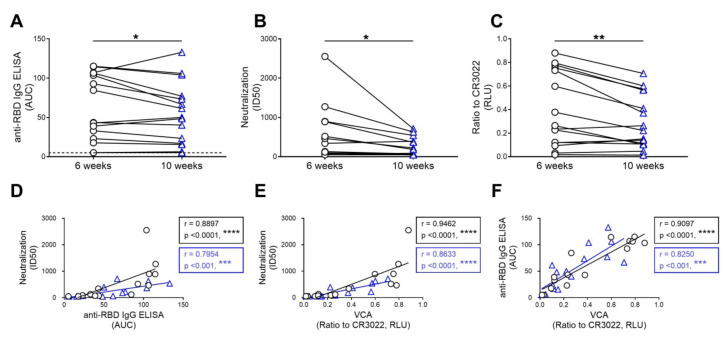
The capacity of convalescent plasma to bind to the S glycoprotein of SARS-CoV-2 and neutralize pseudoviral particles decreases over time. Convalescent plasma samples recovered six (black circle) and ten (blue triangle) weeks after symptoms onset from fifteen individuals were used for (**A**) anti-SARS-CoV-2 RBD IgG ELISA, (**B**) neutralization against pseudoviral particles expressing the SARS-CoV-2 S glycoprotein, and (**C**) virus capture assay (VCA). (**A**) Anti-RBD antibody binding was detected using anti-IgG-HRP. Relative light units (RLU) obtained with BSA (negative control) were subtracted and further normalized to the signal obtained with the anti-RBD CR3022 mAb present in each plate. The areas under the curve (AUC) were calculated based on RLU using GraphPad Prism software. AUC below threshold (dashed line) was considered no binding. (**B**) Neutralization half-maximal inhibitory plasma dilution (ID_50_) was determined using a normalized non-linear regression using Graphpad Prism software. (**C**) VSV-G-pseudotyped viral particles expressing the SARS-CoV-2 S glycoprotein were added to plates coated with plasma samples. Free virions were washed away, and Cf2Th cells were added to the wells. After 48 hours, cells were lysed, and the luciferase activity was measured. Luciferase signals were normalized to those obtained with the RBD-specific CR3022 antibody. Data shown are the mean ± SD of three independent experiments performed in triplicate. Statistical significance was evaluated using a paired t-test (*, *p* < 0.05, **, *p* < 0.01). Correlations between (**D**) neutralization and anti-RBD IgG ELISA, (**E**) neutralization and VCA, and (**F**) anti-RBD IgG ELISA and VCA were calculated. Correlations for samples recovered six weeks (in black) and ten weeks (shown in blue) post symptoms onset are shown. Statistical significance was established with the Spearman rank correlation test (***, *p* < 0.001, ****, *p* < 0.0001).

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
