# Peer review of "Antibody Binding to SARS-CoV-2 S Glycoprotein Correlates with but Does Not Predict Neutralization"

_viruses, 2020, doi:10.3390/v12111214_

Round 1

Reviewer 1 Report

Authors correctly replied to my previous criticisms and suggestions

Reviewer 2 Report

In the revised version of the manuscript, the authors provide additional ELISA data that correlate with their VCA. The revised version is now of higher value for its readers, is written clearer in certain aspects and thus I recommend to accept it its current version.

This manuscript is a resubmission of an earlier submission. The following is a list of the peer review reports and author responses from that submission.

Round 1

Reviewer 1 Report

The article by Ding et al investigates the antibody binding to SARS-CoV-2 S glycoprotein 3. Authors report correlation with neutralization; however, it didn't predict neutralisation. Although this is an interesting study, I have serious concerns about the methodology and inference drawn from these experiments. The topic is of great clinical interest but the methodology lacks rigour. Authors do not state what statistical methods they use to investigate the association and whether baseline clinical factors such as the baseline severity of the infection, virulence and other clinically relevant variables such as comorbidities were accounted/adjusted for in the final model. Recent studies have demonstrated that miRNAs targeting COVID-19 decrease with ageing and underlying conditions. 

Authors also need to incorporate the limitations of the viral capture essay. It is not clear if these results were complemented or replicated with another assay/method. The sensitivity and specificity of this assay in this study haven't been reported nor their association with the stage of their infection. 

Reviewer 2 Report

Very interesting paper particularly for the the methodology conducted with plasma and antibodies. Regarding the humoral response it may be understandable, not all immunological responses are identical for the various polymorphisms present; it could depend on which epitopes of Spike are recognized. It is likely that, in the percentage of cases in which the binding alone with the antibody does not neutralize but binds Spike, the neutralization of the virus occurs with the co-participation of the cellular response. Another consideration is that in these individuals, in addition to antibodies to Spike, there are also antibodies to other viral proteins that cooperate in neutralization. This explains, in my opinion, that individuals with binding but non-neutralizing antibodies still recover and therefore the virus does not spread. Authors should discuss these issues in the discussion. For example, it is possible to do the same type of experiments with spike and another viral protein in association, to understand if in these cases the neutralizing antigen combination? I understand that this takes more time, but it would still be useful for this to be hypothesized and hoped for in the discussion.